# Sperm Nuclei Analysis and Nuclear Organization of a Fertile Boar–Pig Hybrid by 2D FISH on Both Total and Motile Sperm Fractions

**DOI:** 10.3390/ani11030738

**Published:** 2021-03-08

**Authors:** Viviana Genualdo, Federica Turri, Flavia Pizzi, Bianca Castiglioni, Donata Marletta, Alessandra Iannuzzi

**Affiliations:** 1Institute for Animal Production System in Mediterranean Environment, National Research Council, Portici, 80055 Napoli, Italy; viviana.genualdo@cnr.it; 2Institute of Agricultural Biology and Biotechnology, National Research Council, 26900 Lodi, Italy; federica.turri@ibba.cnr.it (F.T.); flavia.pizzi@ibba.cnr.it (F.P.); bianca.castiglioni@ibba.cnr.it (B.C.); 3Department of Agriculture, Food and Environment, University of Catania, 95131 Catania, Italy; d.marletta@unict.it

**Keywords:** fertile pig hybrid, meiotic segregation, sperm FISH analysis, Percoll gradient, nuclear organization

## Abstract

**Simple Summary:**

Mammalian hybrids frequently form in nature between closely related species, and the majority are sterile due to the production of chromosomally unbalanced gametes. In Italy, the widespread wild boar has had negative consequences for free-range pig farming, which is considered the best practice for pig welfare and is a common method of farming most autochthonous pig breeds. This study aimed to analyze the sperm DNA integrity, sperm meiotic segregation and nuclear spatial organization in a boar–pig hybrid, for the first time, to evaluate its fertilizing capacity. The results show that the hybrid presented a high frequency (64%) of motile spermatozoa with a regular chromosome composition and a specific spatial distribution. This study underlines how fertile boar–pig hybrids represent a growing problem for conserving autochthonous pig breeds.

**Abstract:**

A wide range of mammalian hybrids has recently been found by chance or through population-screening programs, but studies about their fertilizing capacity remain scarce and incomplete. Most of them are assumed to be sterile due to meiotic arrest caused by the failure of chromosome pairings. In this study, we evaluated both sperm meiotic segregation, by 2D fluorescent in situ hybridization (FISH) analysis, and sperm quality (Sperm Chromatin Structure Assay) by flow cytometer in a fertile boar–pig hybrid (2n = 37,XY) originating from a Nero Siciliano pig breed (*Sus scrofa domesticus*) and a wild boar (*Sus scrofa ferus*). Spermatozoa were also separated by a dual-layer (75–60%) discontinuous Percoll gradient, resulting in two fractions with a significantly better overall quality in the motile sperm fraction. These data were confirmed by FISH analysis also, where the frequencies of spermatozoa with a regular chromosome composition were 27% in total sperm fraction and 64% in motile sperm fraction. We also evaluated the nuclear architecture in all counted spermatozoa, showing a chromatin distribution changing when chromosome abnormalities occur. Our results demonstrate that the chromosome pairing has a minimal effect on the sperm segregation and semen quality of a boar–pig hybrid, making it fertile and harmful for the conservation of autochthonous pig breeds.

## 1. Introduction

Over the last decade, several studies have reported a rising number of mammalian hybrids, originating by accident or design [1,2,3] and considered sterile with some exceptions [4]. Recent studies showed crosses between domestic pig (DP) and wild boar (WB), belonging to the same *Sus scrofa* (*Ssc*) species [5]. The hybridization between DP (*Sus scrofa domesticus*, 2n = 38,XX/XY) and WB (*Sus scrofa ferus,* 2n = 36,XX/XY) seems to be very frequent, especially when they share the same area, grazing on high-altitude pastures and in forests [6]. There are very few studies about boar–pig hybrid fertility, and the existing studies are mainly related to the fertility of sows, which is considered the primary source of economic loss for pork producers [7].

With the introduction of assisted reproductive technology (ART), advanced techniques such as flow cytometry (FCM) have been applied in several studies to provide accurate and unbiased evaluation of sperm functions, such as sperm DNA integrity [8,9,10]. Sperm DNA damage and chromatin structure alterations extrapolated from the Sperm Chromatin Structure Assay (SCSA) are considered the most valuable parameters for the assessment of male fertility [8]. Furthermore, fluorescent in situ hybridization (FISH) analysis gives an estimation of the frequency of chromosomal abnormalities (CA) by measuring sperm aneuploid (deviation from the normal haploid or diploid state). It is widely shown that gametes of subfertile individuals tend to show higher rates of CA than fertile ones. In fact, CA modify the meiotic division of germ cells, resulting in normal and translocated (unbalanced spermatozoa) chromosome formation by two types of segregation: 2:2 (alternate, adjacent I or adjacent II) and 3:1. This condition can cause a high percentage of chromosomally unbalanced gametes, reducing the reproductive success of an individual and increasing their risk of transmitting CA to offspring [11].

WB is characterized by both high reproductive potential (due to the untimely puberty) [12] and a high rate of disease transmission to farms [13], leading to a genetic introgression by hybridization of DP and WB populations. In this way, more adaptive genotypes can be generated in the environment, inducing a local increase of new invasive species with severe demographic impacts, to the detriment of autochthonous breeds [14]. For this reason, the European Commission (EC) aims to preserve the purebred animals registered in the herdbook, avoiding the reproduction of hybrid-derived invasive species. In the last few years, about one million WB have populated Italy [15], leading to negative consequences for free-range pig farming, the best practice for pig welfare and a common Italian method for farming several autochthonous pig breeds [16], such as Nero Siciliano (NS).

The NS is farmed in the Nebrodi Mountains (in the northeast of Sicily), and its population is increasing (6629 registered in 2020 vs. 900 recorded in 2001), according to Domestic Animal Diversity Information System (DAD-IS) data (http://www.fao.org/dad-is/en/, accessed on 28 February 2021) and the herdbook of National Swine Breeders’ Association (ANAS). NS shows phenotypic resemblance to WB but has a smaller size (65 cm at the withers for an adult male), a darker coat and earlier sexual maturity. Moreover, it is a long-lived breed, disease-resistant and able to convert natural graze or poor food (roughage) into high-quality products (meat) [17]. Since the 1990s, the EC has been conserving some local breeds through the GENRES project “European gene banking project for the pig genetic resources”. In this way, the genetic material of NS and other local Italian pig breeds such as Mora Romagnola, Casertana and Cinta Senese was collected and stored in the “Semen Bank of Italian local pig breeds” [18,19].

In this study, we have analyzed the sperm DNA integrity, sperm segregation, aneuploidy and nuclear organization of one fertile boar–pig hybrid in both total (TSF) and motile (MSF) sperm fractions for the first time.

## 2. Materials and Methods

### 2.1. Animals and Sampling

During cytogenetic screening of 81 endangered local pig breeds, two NS boars (from the same farm) were found to have resulted in a boar–pig hybrid 2n = 37,XY rob(15;17) [20]. The two animals were casually detected, considering their phenotype was not distinguishable from that of the investigated pig breed. They turned out to be brothers, having the same father. The two hybrids were registered in ANAS’s herdbook as fertile and had generated registered offspring. The semen samples were collected by an epididymal sperm extraction technique immediately after surgical castration at the farm. The standard semen collection procedure by artificial vagina was not applicable because the boars were unaccustomed to human contact, having been farmed free-range. Spermatozoa were extracted by epididymis using the retrograde flushing technique, frozen on liquid nitrogen vapor and transferred to IBBA Laboratories in Lodi (northern Italy) for semen quality analysis and storage.

The evaluation of the quality and fertilizing ability of semen is an essential aspect in guaranteeing its “conservation value” and storing it in animal cryobanks. In this analysis, we included both a boar–pig hybrid 2n = 37,XY rob(15;17) and a control DP 2n = 38,XY.

### 2.2. Freezing Protocol of Epididymal Spermatozoa

The epididymal semen samples were diluted using a two-step procedure: the first 50% was added (Extender Boarciphos A (IMV Technologies, L’Aigle, France)), followed by the second half (Boarciphos B (IMV Technologies, L’Aigle, France)). In detail, sperm cells were flushed in a retrograde direction from the ductus deferens through the cauda epididymidis with a syringe loaded with approximately 4 mL of warmed (37 °C) Boarciphos A extender (IMV Technologies, L’Aigle, France), supplemented with 20% of egg yolk (*v/v*). Sperm cells collected were equilibrated for 90 min at 4 °C. After equilibration, sperm cells were diluted in the freezing medium Boarciphos B supplemented with 10% glycerol, 20% egg yolk and 2% Equex STM paste (IMV Technologies, L’Aigle, France) (*v*/*v*), to achieve the final concentration of 500 × 10^6^ sperm/mL that was then packed into 0.5 mL CBS High Security Straws (CryoBioSystem, Paris, France) and sealed using polyvinyl alcohol powder. Semen straws were frozen in nitrogen vapor by placing on a rack at 7 cm above the nitrogen level for 15 min. Subsequently, straws were plunged into liquid nitrogen and stored at −196 °C in a liquid nitrogen tank.

### 2.3. Percoll Gradient

Frozen–thawed spermatozoa were separated through a dual-layer (75–60%) discontinuous Percoll gradient, following the protocol of Suzuki et al. with some modifications [21]. For each boar (hybrid and control), two frozen semen doses of 0.5 mL, containing approximately 2.5 × 10^6^ spermatozoa, were simultaneously thawed in a water bath at 42 °C for 20 s and pooled. The pool (1 mL) was split into two aliquots of 500 μL. One aliquot was evaluated as TSF. The second was subjected to a Percoll gradient to obtain an MSF, considered the highest-quality semen fraction. Then, 500 µL of the thawed semen (for each boar) was washed by centrifugation at 600× *g* for 8 min in 5 mL of a saline solution for washing spermatozoa (WS-PVA), consisting of 140 mM NaCl, 20 mM HEPES and 1 mg/mL polyvinyl alcohol (PVA), adjusted to pH 7.4.

The supernatant was discarded, and pelleted spermatozoa were resuspended in 1 mL of WS-PVA. It was overlaid on a two-step Percoll with the upper part of 60% and lower part of 75% concentrations in two 15 mL conical tubes from each boar. The tubes were centrifuged at 200× *g* for 5 min. After the centrifugation, the MSF obtained was resuspended in up to 3 mL of WS-PVA for washing centrifugation (900× *g* for 15 min). Supernatants were discarded and pelleted spermatozoa were resuspended in up to 3 mL of WS-PVA and washed twice by centrifugation (600× *g* for 8 min).

### 2.4. Sperm DNA Integrity

Sperm DNA fragmentation and chromatin condensation were assessed using the SCSA test by Evenson et al. [22] and further validated on our Guava easyCyte 5HT flow cytometer according to Sellem et al. [23]. The fluorescent probes were excited by a 20 mW argon ion laser (488 nm). We analyzed 5000 spermatozoa events per sample, for each boar, at a flow rate of 200 cells/s. The SCSA test was performed as previously described by other studies [24,25]. Briefly, 3.0 × 10^5^ spermatozoa were diluted in 200 μL of TNE buffer (0.01 M Tris-HCl, 0.15 M NaCl, 1 mM ethylenediaminetetraacetic acid (EDTA), pH 7.4) and added to 400 μL of an acidic solution (Triton X−100 0.1%, 0.15 M NaCl, 0.08 M HCl; pH 1.2). After 30 s, cells were stained with 1.2 mL of acridine orange solution (6 μg/mL in 0.1 M citric acid, 0.2 M Na_2_HPO_4_, 1 mM EDTA, 0.15 M NaCl; pH 6). After 2.5 min, two replicates per sample were read on a flow cytometer. We quantified the type of DNA damage with the following parameters: (1) the extent of DNA denaturation, expressed as DNA fragmentation index (%DFI), which indicates the percentage of sperm cells outside the main sperm population with fragmented DNA; (2) the percentage of immature cells with reduced nuclear condensation (incomplete histone–protamine exchange), expressed as a percentage of sperm with high green (%HG) fluorescence.

Data were acquired and analyzed using cytoSoft and IMV easySoft software (Merck KGaA, Darmstadt, Germany; distributed by IMV Technologies), respectively. The latter created a %DFI and %HG histogram profile of the entire sperm population. Computer gates of the %DFI and %HG histogram were used to determine the %DFI and %HG.

### 2.5. Slide Preparation and Decondensation

Preparation and decondensation of spermatozoa followed the protocol of Di Dio et al. [11] with some modifications. The semen of each animal, obtained from both TSF and MSF fractions, was washed in 6 mmol/L EDTA and fixed in 3:1 methanol/acetic acid. Four different kinds of samples were prepared: (1) TSF from the hybrid, (2) MSF from the hybrid, (3) TSF from the control and (4) MSF from the control. Sperm DNA was denatured by immersion in 3 mmol/L NaOH at room temperature. The optimal decondensation time was determined experimentally.

### 2.6. Probe Preparation and Sperm FISH Analysis

Each probe used in the study consisted of a pool of three bacterial artificial chromosomes (BACs) selected in a contig (maximal distance between clones: 400 Kb) to increase the size of the sequence covered by the probe (around 1 Mb) and thus the intensity of the FISH signals (Table 1). We obtained them from the CHORI-BACPAC Resources Center (CH-242 library). We first tested them on *Ssc* lymphocyte R-banded metaphases to check their physical localization. The schematic representation of the translocation and localization of the BACs used in this study is illustrated in Table 1. DNA isolation was performed using protocol recommended by Children’s Hospital Oakland Research Institute (CHORI) [26]. The preparation of probes and hybridization by dual-color FISH analysis followed the protocols described by Iannuzzi et al. [27].

### 2.7. Statistical Analysis

The chi-square test [28] was used to compare (1) the frequencies between Percoll and whole semen selection methods within each boar, (2) the frequencies between hybrid and control boar within each sperm selection method and (3) the alternate standard and rob segregation sperm frequencies of the hybrid. A *p*-value of less than 0.001 was considered significant.

Data obtained from SCSA assessment were analyzed using the SAS package v 9.4 (SAS Institute Inc., Cary, NC, USA). The general linear model procedure (PROC GLM) was used to evaluate the Percoll separation’s efficiency on the SCSA sperm quality parameters. The model included fixed effects of the fraction and the animal. Results are given as adjusted least squares means ± standard error means (LSM ± SEM). We set the level of significance at *p* < 0.05.

## 3. Results

### 3.1. Sperm DNA Integrity

Results relative to the sperm chromatin structure are shown in Table 2. We found some differences in DNA integrity within the sperm fractions (TSF and MSF) and between the two animals.

Sperm cells were successfully separated through the dual-layer (75–60%) discontinuous Percoll gradient, showing a higher subjective motility and lower %DFI in the motile sperm fraction. Considering the percentage of %DFI, a reduction in the DNA damage was noticed in the MSF fraction in both animals, and this reduction was significant in the hybrid sample. Hybrid boar showed higher values of %HG, representative of the percentage of immature spermatozoa with reduced nuclear condensation.

### 3.2. FISH Analysis

Figure 1 shows the predicted segregation patterns of *Ssc* chromosomes 15, 17 and rob(15;17). Segregation analysis was achieved on 10,000 spermatozoa (evaluated in steps of 2000 and 5000) for TSF and MSF of the hybrid and 2000 for TSF and MSF of the control samples (Table 3). The efficiency of hybridization was about 99.8%, where only well-decondensed spermatozoa heads exhibiting high-intensity signals were considered for the analysis. The meiotic segregation pattern for the carrier was determined by double-colored FISH signals corresponding to *Ssc* chromosomes 15 (green signal) and 17 (red signal), as shown in Figure 2. To distinguish alternate normal (15 and 17 *Ssc* chromosomes) from rob(15;17) spermatozoa, we considered all those that presented two detached signals as normal (Figure 2A) and all those that showed the green and red (attacked) or yellow signals as rob(15;17) carriers (Figure 2B,C).

Furthermore, we evaluated the nuclear architecture of these chromosomes in all counted spermatozoa in specific regions: the green (*Ssc* 15) detached signal was detected in the apical part of the sperm head (85%), the red (*Ssc* 17) detached signal was detected in the medial part of the sperm head (85%) and the green and red (attacked) or yellow signals were detected in the basal part of the sperm head (88%). We evaluated them in the control, finding the same localization for *Ssc* 15 and 17 chromosomes (96%) as shown in Figure 3. We revealed that 27% of hybrid TSF resulted as normal (compared to control with 98%), while 26% carried the rob(15;17) (in contrast to control with 2%). Moreover, 64% of hybrid MSF was normal (in contrast to control with 97%), whereas 23% carried the rob(15;17) (in contrast to control with 2%). However, we found a higher percentage of adjacent I (32%) and a smaller percentage of adjacent II (11%) in TSF of the hybrid, not detected in MSF.

## 4. Discussion

The bulk of sperm DNA fragmentation studies suggest that sperm with damaged DNA will fertilize eggs [29]. In the literature, the estimated threshold above which the percentage on the DNA fragmentation index (%DFI), from the SCSA test, negatively impacts fertility varies across species; e.g., this threshold is 6% for pigs, 10–20% for bulls, 28% for horses and 25–30% for humans [10]. In the present study, the proportion of sperm DNA damage was found to be considerably low (mean %DFI = 1.74) in frozen–thawed boar semen, placing below the threshold %DFI for subfertility. This confirms the limited vulnerability of DNA to freezing and thawing compared with other sperm cell components, such as plasma membranes and mitochondria [30]. The fragmented levels of DNA fragmentation index (%DFI) of our study agree with other studies on the SCSA outcome of frozen–thawed boar semen, where the mean %DFI was 2.60 and 2.1 [31,32].

We analyzed the sperm segregation in both TSF and MSF, mainly to examine the Percoll gradient technique’s efficiency in eliminating spermatozoa with an abnormal chromosomal constitution, reporting the total percentage of high-quality spermatozoa able to fertilize. Percoll separation methods (colloidal suspension of polyvinyl pyrrolidone coated silica particles) have been broadly used for in vitro fertilization (IVF) and have been reported to have a positive correlation with sperm quality (induced by motility, viability, normal morphology and fertility) [33,34]. In pigs, spermatozoa separated by Percoll gradient from fresh or frozen–thawed semen presented high motility, intact plasma and acrosomal membranes [35,36]. In this study, the MSF was successfully separated through the dual-layer (75–60%) discontinuous Percoll gradient, showing a significant reduction in the DNA damage considering the %DFI variables, as demonstrated by Larson et al. [37]. The Percoll gradient enriched sperm chromatin integrity, as observed when spermatozoa were assessed using SCSA. Higher percentages of immature spermatozoa, with reduced nuclear condensation (%HG), were observed in the hybrid boar due to the meiotic problems (higher number of diploid spermatozoa). This parameter might be useable to discriminate between particular subjects, including in cytogenetic analyses as suggested by Nagy et al. [38].

Moreover, we found a positive correlation with alternate sperm segregations, proving that this method effectively reduced the percentage of spermatozoa carrying unbalanced chromosomal aberrations [39]. Of the boar–pig hybrid spermatozoa selected by Percoll (MSF), 67% were normal, compared to 27% of those detected in the TSF (Table 3 and Table 4). On the other hand, we did not find different percentages of spermatozoa carrying the rob(15;17), indicative of the hybrid genotype, in either selected method. This could be due to their balanced chromosomal constitution, which has probably no impact on the sperm motility selected by the Percoll gradient, as reported by Vozdova et al. [39].

We also analyzed the chromosome positioning of *Ssc* 15, 17 and rob(15;17) in all sperm nuclei, not only to evaluate their nonrandom position but also to study sperm intranuclear localization chromosomes involved in a rob. These data are comparable with the results shown in pigs by Ducos et al. [40] and Rubes et al. [41].

The spermatozoa are highly differentiated cells resulting from the several-step process of spermatogenesis. During the stages following meiosis, chromatin undergoes structural reorganization, and DNA becomes very condensed and genetically inactive [42]. The spatial arrangement of chromosomes during prometaphase, metaphase and interphase seems nonrandom [43,44,45]. Several studies have demonstrated specific and nonrandom chromosome architecture in mammalian sperm nuclei. Foster et al. [46] studied genome organization by chromosome positioning in pig sperm nuclei using the 2D-FISH analyses for the evaluation of both autosomes (*Ssc* 1–18) and the sexual chromosome (X–Y) in a normal segregation. They demonstrated that all chromosomes are organized in specific chromosome territories, tracing *Ssc* 15 in the apical-medial and *Ssc* 17 in the medial-basal part of spermatozoa. We found the same medial localization of the *Ssc* 17 chromosome, but we localized *Ssc* 15 in the apical region (referring to spermatozoa with normal chromosome segregation). One probable reason could be the type of probe used for the analysis. Foster et al. used probes that painted the whole chromosome, producing larger signals due to the detection of entire chromosomes (20–30 Mb), while we used a pool of Bac probes with a smaller signal (1 Mb).

Acloque et al. [47] compared the global architecture of boars’ sperm nuclei with both regular and a rob(13;17) segregation by 3D sperm FISH analyses. They found a medial and apical localization for *Ssc* 17 and *Ssc* 13, confirming our result for *Ssc* 17 in normal spermatozoa. Furthermore, they revealed a different localization of spermatozoa carrying the rob(13;17), probably induced by the fusion between the involved chromosomes. In our case also, we found a different localization (basal) of spermatozoa carrying both the rob(15;17) and any other abnormal segregation detected (Figure 2 and Figure 3). In this way, we can speculate that spermatozoa carrying a chromosomal translocation (CT) may change spatial chromosome distribution and probably reduce their quality and motility, as demonstrated by their lower percentage detected in MSF. Several studies have shown how CT can change sperm intranuclear localization, influencing the other chromosome localization [48,49], but have not provided information about the vitality and motility of abnormal sperm. In this study, we have positively correlated the DNA integrity with both normal segregation and spatial distribution. We have also demonstrated that the hybrid could generate offspring with 67% normal spermatozoa and a registered offspring.

## 5. Conclusions

Biotechnologies such as the Percoll gradient, flow cytometry evaluation, FISH analysis and a pool of BAC probes have been applied together to obtain a more precise analysis of DNA integrity and sperm segregation.

In this study, we have achieved the essential objectives of (a) studying the sperm segregation in a fertile hybrid, reporting also the percentage of each sperm fraction; (b) showing the correlation between the high-quality sperm fraction (selected by Percoll gradient according to DNA integrity) and a lower sperm aneuploidy percentage; and (c) showing the modification of nonrandom chromosome spatial organization in sperms carrying a CA.

Finally, thanks to the cytogenetic monitoring of local pig breeds, CA can be easily detected, preserving the reproducers’ genetic selection.

Further studies are needed for a better understanding of the correlation among motility, chromatin integrity and spatial chromosome distribution when a CA occurs.

## Figures and Tables

**Figure 1 animals-11-00738-f001:**
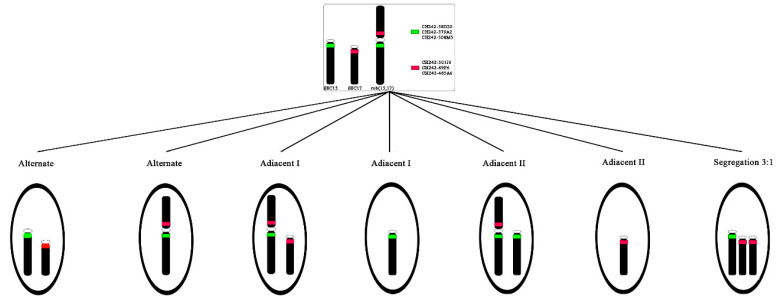
Illustration of the different gametes produced by 2:2 and 3:1 segregation mechanism of *Ssc* 15 and 17 chromosomes involved in the rob, with localization of the DNA probes.

**Figure 2 animals-11-00738-f002:**
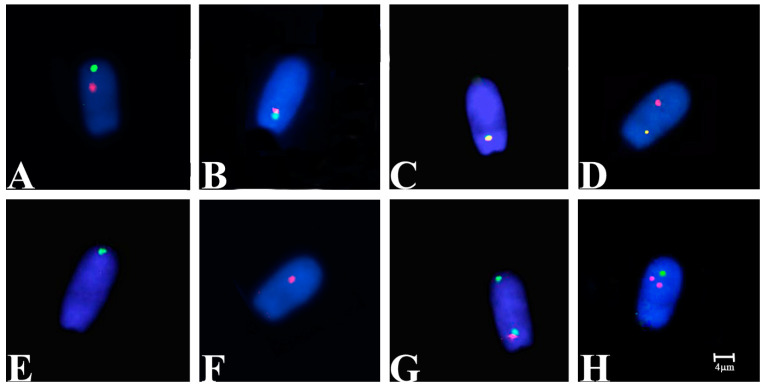
Representative spermatozoa heads after FISH analysis with 1st (*Ssc* 15) and 2nd (*Ssc* 17) pools stained in G (green) and R (red), respectively. (**A**) Alternate normal sperm nucleus with separate G and R signals. (**B**,**C**) rob(15;17)-carrier sperm nucleus with attacked G and R or Y (overlapped) signals. (**D**) Adjacent I sperm nucleus with R and Y signals. (**E**) Adjacent I sperm nucleus with G signal. (**F**) Adjacent II sperm nucleus with R signal. (**G**) Adjacent II sperm nucleus with G and RG signals. (**H**) A 3:1 sperm nucleus with double R and single G signals.

**Figure 3 animals-11-00738-f003:**
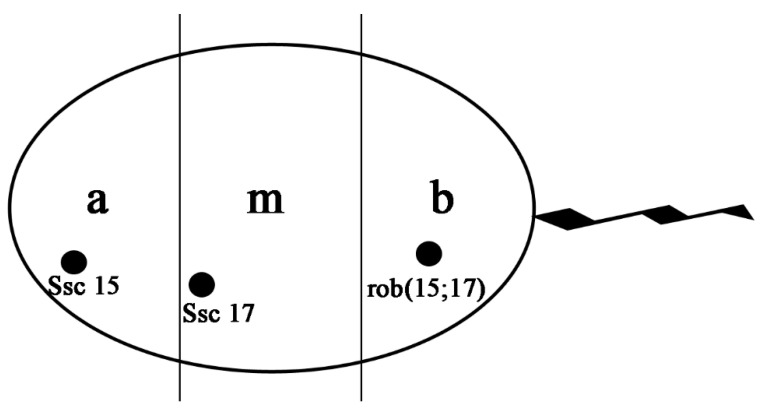
Schematic representation of porcine sperm nucleus showing the borderline anterior (a), medial (m) and basal (b) sections with the *Ssc* 15, *Ssc* 17 and rob(15;17) spatial localizations.

**Table 1 animals-11-00738-t001:** Bacterial artificial chromosome (BAC) probes with their genome view (according to NCBI) used for FISH analysis.

Probe	*Ssc* CH-242 Genome Localization (bp)	Chromosomal Localization	Label
**1st Pool**			
bI 0038D20	3,378,177–3,735,904		
bI 00379A02	4,104,594–4,360,253	*Ssc* 15q12	Biotin
bI 0508M05	4,664,773–4,976,019		
**2nd Pool**			
bI 0501J06	4,740,641–5,086,289		
bI 0069P06	5,160,276–5,425,104	*Ssc* 17q12	Digoxigenin
bI 0465A06	5,711,917–5,976,153		

**Table 2 animals-11-00738-t002:** Results of Sperm Chromatin Structure Assay obtained from flow cytometry analysis, for animal and sperm fraction.

Variables	Control		Hybrid	
	TSF	MSF	TSF	MSF
%DFI (%)	2.03 ± 0.3437 ^aA^	1.58 ± 0.3437 ^aA^	2.51 ± 0.3437 ^aA^	0.87 ± 0.3437 ^bA^
%HG (%)	0.11 ± 0.0268 ^aA^	0.20 ± 0.0268 ^bA^	0.34 ± 0.0268 ^aB^	0.90 ± 0.0268 ^bB^

TSF: total sperm fraction; MSF: motile sperm fraction; %DFI: fragmented DNA sperm; %HG: high green fluorescence sperm. Different superscripts in the same row correspond to a significant difference (*p* < 0.05) between fraction (within animal, lowercase letter) and animal (within sperm fraction, uppercase letter).

**Table 3 animals-11-00738-t003:** Comparison of the meiotic segregation on both rob(15;17) carrier and the control, detected on TSF and MSF. The G-R refers to green and red detached signals (*Ssc* 15 and 17); the GR or Y refer to green and red (attacked) or yellow overlapped signals, indicative of rob(15;17).

Fluorescent Signals ^a^	Segregation	Chromosomal Constitution	Associated Genotype	% Of Combinations (Number of Gametes Investigated)			
				Hybrid TSF	Control TSF	Hybrid MSF	Control MSF
G-R	Alternate	*Ssc* 15-17	n = 19	26.73 (2673)	96.05 (1921)	66.63 (6663)	96.70 (1934)
				49.61 ^b^	97.96 ^b^	74.16 ^b^	98.17 ^b^
GR		rob(15;17)	n = 18	23.97 (2397)	2 (40)	20.21 (2021)	1.25 (25)
				44.48 ^b^	2 ^b^	22.49 ^b^	1.26 ^b^
Y		rob(15;17)	n = 18	3.18 (318)	0 (0)	3.00 (301)	0.55 (11)
				5.90 ^b^	0 ^b^	3.35 ^b^	0.55 ^b^
		tot		53.88 (5388)	98.05 (1961)	89.85 (8987)	98.50 (1970)
R-GR (Y)	Adjacent I	*Ssc* 17; rob(15;17)	n = 18	21.98 (2198)	0.90 (18)	2.040 (204)	0.85 (17)
G		*Ssc* 15	n = 19	10.10 (1010)	0 (0)	4.10 (401)	0 (0)
		tot		32.08 (3208)	0.90 (18)	6.04 (605)	0.85 (17)
R	Adjacent II	*Ssc* 17	n = 19	9.97 (997)	0 (0)	4.10 (410)	0.65 (13)
G-GR		*Ssc* 15; rob(15;17)	n = 18	1.03 (103)	1.05 (21)	0 (0)	0 (0)
		tot		11.0 (1100)	1.05 (21)	4.09 (410)	0.65 (13)
G-R-R	3:1	*Ssc* 15; *Ssc* 17; *SSC* 17	n = 19	3.04 (304)	0 (0)	0 (0)	0 (0)
Total				10,000	2000	10,000	2000

^a^ Fluorescence signals detected on decondensed sperm nuclei: G = green; R = red; Y = yellow. ^b^ Percentage refers to alternate segregation total count sperms.

**Table 4 animals-11-00738-t004:** Alternate sperm segregation of rob(15;17) carrier by chi-square test.

Alternate Segregations	TSF n (%)	MSF n (%)
Normal (G-R)	2673 (50) ^A^	6663 (74) ^B^
rob(15;17) (GR or Y)	2715 (50) ^A^	2443 (26) ^B^
Total	5388	8985

^A,B^ Values with different superscripts within rows are different (*p* < 0.001).

## Data Availability

Data is contained within the article.

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
