# Peer review of "Sperm Nuclei Analysis and Nuclear Organization of a Fertile Boar–Pig Hybrid by 2D FISH on Both Total and Motile Sperm Fractions"

_animals, 2021, doi:10.3390/ani11030738_

Round 1

Reviewer 1 Report

The Authors made the necessary  improvements according to my suggestion. I have no other objections. In my opinion the manuscript can be published in ‘Animals’.

Reviewer 2 Report

Despite having evaluated a single animal, which prevents something from being said regarding the fertility of the hybrid animal population, the article brings an interesting approach.

Here are considerations:

What is the semen freezing method used?

In the introduction, the article should clarify the correlation between the evaluations carried out and fertility.

Line 99 - Who are the control animals?

Item 2.3 - better explain how the evaluated parameters and the meaning of each one were calculated.

Item 3.3 - because there is an item of statistical analysis within the results

Table 2 - which letters represent the comparison between the lines? And between the columns? This should be described at the bottom of the table.

Line 250 - How (what is the mechanism) would the Percoll gradient separate the sperm according to their DNA damage?

Line 251 - How was the correlation analysis performed? It is not described in MM.

Improve the conclusion. The missing link between completion and objective

Reviewer 3 Report

The present manuscript deals with an important and interesting topic, which fits well to the scopes of the Journal. I found the cytogenetic part of the study elegant and sound; the spermatology part, however, is rather weakly written and the interpretation of the results is not correct, therefore a major revision is needed.

The SCSA parameter nomenclature is rather mixed, old terminology (alpha-t) is used together with the correct, new terminology (DFI). Please refer to the relevant articles of DP Evenson about the terminology change (for ex., Animal Reproduction Science 169 (2016) 56–75 - even Dr. Evenson admits that there is some confusion with the terminologies, but I would suggest to stick to the new terms).

Authors should provide more detailed information about the freezing of the sperm samples. What extender was used? Were the samples frozen in straws (what size?) or flat packs? This part needs an extensive re-writing and more details are needed to explain the sperm freezing protocol.

The subpopulation after Percoll treatment should not be called Motile Sperm Fraction as motility was not evaluated in this study (or at least not included in the manuscript).

The reduction of the extent of sperm DNA damage after sperm cleansing is not a new finding, Authors should refer to previous publications (for ex., Hallap et al., Theriogenology, 2005 Apr 1;63(6):1752-63).

Hybrid boar showed larger High Green Flurescence values, as it is presented in table 2. It was already suggested before, that this parameter may be useful to identify animals with cytogenetic abnormalities (Nagy et al, Theriogenology, 79 (2013) 1153–1161).

In the Discussion, Authors compare their SCSA findings to the results on other species. Here they should rather compare their results to other SCSA studies on boar sperm, for ex.: Didion et al., Boar fertility and sperm chromatin structure status: a retrospective report. J Androl. 2009 Nov-Dec;30(6):655-60. or Hernández et al., Differences in SCSA outcome among boars with different sperm freezability. Int J Androl. 2006 Dec;29(6):583-91. but there are even more studies on boar sperm DNA integrity.

"we have proved the limited vulnerability of DNA to freezing and thawing, compared with other sperm cell components, such as plasma membranes and mitochondria" - this is well known, not a new finding.

"In this study, we have positively correlated the sperm quality with both normal segregation and spatial distribution." actually, sperm quality was not fully checked in the present study, only one component, DNA integrity.

Minor comment: please use "spermatozoa" instead of "sperms" throughout the manuscript.

Round 2

Reviewer 3 Report

Authors have addressed all comments and concerns mentioned in my previous review, so in my opinion the manuscript can be accepted for publication. Congratulations for this very elegant study!

This manuscript is a resubmission of an earlier submission. The following is a list of the peer review reports and author responses from that submission.

Round 1

Reviewer 1 Report

Review Manuscript ID: animals--1031752, entitled ‘Sperm Nuclei Analysis and Nuclear Organization of a Fertile Boar-Pig Hybrid by Fish Analysis on both Total and Motile Sperm Fractions’

Comments and Suggestions for Authors

The article entitled: “Sperm Nuclei Analysis and Nuclear Organization of a Fertile Boar-Pig Hybrid by Fish Analysis on both Total and Motile Sperm Fractions” addresses an interesting topic. However, the study design is not clear, methods have serious flaws.

Line 29-30: Incomprehensible sentence. Because in the previous sentence you say that research into the fertility of mammalian hybrids are scarse and incomplete.

Line 90: „To date,…….” Too bold thesis. What did the Authors mean?

Materials and Methods

Line 96-105: Semen frozen from two boars was used. There are many studies on the negative effects of freezing boar semen. Why wasn't fresh semen used in the research? The freezing point can damage boar sperm.

What mean “satisfactory semen quality”? How was it checked?

Line 100: Why did you pooled the semen of two boars? The boar effect?

Line 145: „The semen of each animal, obtained from both entire…… “ – explain.

Line 166-169: too generally

Line 219: Where is Table 4?

249-253: Repeating the aim of paper with the Introduction. Sentence unnecessary.

Line 315: Adapt the references to the requirements of the "Animals"

Reviewer 2 Report

The study focuses on sperm segregation, aneuploidy and nuclear organization of one fertile boar-pig hybrid.  In the current form, the manuscript contains more technical descriptions than real analysis.

The number of samples is too low to draw conclusions about the accuracy of the method and correlations with sperm quality.

Overall, this is a simple technical report (not a research article) that does not provide an accurate picture of the situation.